# A *bvrR/bvrS* Non-Polar *Brucella abortus* Mutant Confirms the Role of the Two-Component System BvrR/BvrS in Virulence and Membrane Integrity

**DOI:** 10.3390/microorganisms11082014

**Published:** 2023-08-05

**Authors:** Olga Rivas-Solano, Kattia Núñez-Montero, Pamela Altamirano-Silva, Nazareth Ruiz-Villalobos, Elías Barquero-Calvo, Edgardo Moreno, Esteban Chaves-Olarte, Caterina Guzmán-Verri

**Affiliations:** 1Centro de Investigación en Biotecnología, Instituto Tecnológico de Costa Rica, Cartago 30109, Costa Rica; 2Laboratorio Facultad Ciencias de la Salud, Instituto de Ciencias Biomédicas, Universidad Autónoma de Chile, Temuco 4813003, Chile; kattia.nunez@uautonoma.cl; 3Centro de Investigación en Enfermedades Tropicales, Universidad de Costa Rica, San Pedro de Montes de Oca, San José 2060, Costa Rica; pamela.altamiranosilva@ucr.ac.cr (P.A.-S.); esteban.chaves@ucr.ac.cr (E.C.-O.); 4Programa de Investigación en Enfermedades Tropicales, Universidad Nacional, Heredia 40104, Costa Rica; nazaret.ruiz.villalobos@una.cr (N.R.-V.); elias.barquero.calvo@una.cr (E.B.-C.); edgardo.moreno.robles@una.cr (E.M.); catguz@una.cr (C.G.-V.)

**Keywords:** *Brucella abortus*, two-component system BvrR/BvrS, attenuated mutants

## Abstract

*Brucella abortus* is a bacterial pathogen causing bovine brucellosis worldwide. This facultative extracellular–intracellular pathogen can be transmitted to humans, leading to a zoonotic disease. The disease remains a public health concern, particularly in regions where livestock farming is present. The two-component regulatory system BvrR/BvrS was described by isolating the attenuated transposition mutants *bvrR*::Tn5 and *bvrS*::Tn5, whose characterization led to the understanding of the role of the system in bacterial survival. However, a phenotypic comparison with deletion mutants has not been performed because their construction has been unsuccessful in brucellae and difficult in phylogenetically related *Rhizobiales* with BvrR/BvrS orthologs. Here, we used an unmarked gene excision strategy to generate a *B. abortus* mutant strain lacking both genes, called *B. abortus ∆bvrRS*. The deletion was verified through PCR, Southern blot, Western blot, Sanger sequencing, and whole-genome sequencing, confirming a clean mutation without further alterations at the genome level. *B. abortus ∆bvrRS* shared attenuated phenotypic traits with both transposition mutants, confirming the role of BvrR/BvrS in pathogenesis and membrane integrity. This *B. abortus ∆bvrRS* with a non-antimicrobial marker is an excellent tool for continuing studies on the role of BvrR/BvrS in the *B. abortus* lifestyle.

## 1. Introduction

*Brucella* organisms are Gram-negative facultative extracellular–intracellular pathogens that cause brucellosis, a neglected zoonotic disease with a worldwide distribution. *Brucella abortus* infects cattle and exhibits a strong tropism for the reproductive system, causing abortion, infertility, decreased milk production, reproductive failure, epididymitis, and hence, economic losses. Humans are accidental hosts, initially developing an acute febrile disease that could become a chronic infection with osteoarticular, gastrointestinal, hepatobiliary, pulmonary, genitourinary, cardiovascular, and neurological complications [1].

Brucellae lack classical virulence factors, and their pathogenicity depends on their ability to invade, survive, and replicate inside host professional and non-professional phagocytes. During their intracellular lifecycle, brucellae avoid the endocytic pathway and redirect their trafficking to a replicative compartment derived from the endoplasmic reticulum [2,3]. Therefore, the adaptation to this intracellular trafficking requires extremely well-coordinated gene expression, mainly through two-component signal transduction systems (TCSs).

In brucellae, the TCS BvrR/BvrS comprises the sensor membrane protein BvrS and its cognate cytoplasmic response regulator BvrR. This TCS was described in 1998 through the isolation and phenotypic characterization of two *B. abortus* kanamycin-resistant transposition mutants, one in *bvrR* and the other in *bvrS*. These mutants were selected based on an association of brucellae virulence with peculiar membrane properties rarely present simultaneously in other Gram-negative pathogens [4], like a low-immunogenic LPS, permeability to hydrophobic agents, and resistance to bactericidal cationic peptides and polymyxin B. Both transposition mutants exhibited an attenuated phenotype characterized by reduced invasiveness and absence of replication in HeLa cells and macrophages, incapability to avoid the lysosomal route, and lack of virulence in the murine model [4]. According to subsequent studies, BvrS probably senses signals associated with the intracellular environment, like low pH and nutrient availability [5,6], and BvrR regulates cell envelope proteins for host–pathogen interactions, among them Omp25 [7], an outer membrane protein with structural functions [8] that is associated with bacterial persistence *in vivo* [9]. BvrR also regulates metabolic fitness [10], pathways for intracellular life [11], and virulence circuits for intracellular traffic and cell egress [5,6].

The use of transposition mutants encoding antimicrobial resistance, although not ideal, has being an alternative to the construction of mutants in this TCS, which has proven difficult to generate in brucellae [4,12] and in other phylogenetically related *Rhizobiales* having orthologs of BvrR/BvrS, like the *Sinorhizobium meliloti* TCS ExoS/ChvI [13,14,15,16,17], the *Agrobacterium tumefacient* TCS ChvG/ChvI [18,19,20], and the *Bartonella henselae* TCS BatR/BatS [21]. Therefore, a clean deletion mutant with non-acquired antibiotic resistance should be obtained to compare and corroborate phenotypes [22]. Here, we constructed a *B. abortus* double-null mutant strain in *bvrR* and *bvrS* called *B. abortus ∆bvrRS*. The phenotype of *B. abortus ∆bvrRS* was attenuated in the cell culture model like the one described for both transposition mutants, validating the role of BvrR/BvrS in virulence.

## 2. Materials and Methods

Bacterial strains, growth conditions, and plasmids. All procedures involving live *B. abortus* were carried out according to the “Reglamento de Bioseguridad de la CCSS 39975–0”, 2012, after the “Decreto Ejecutivo #30965-S”, 2002, and research protocol SIA 0652–19 approved by the National University, Costa Rica. The strains and plasmids used in this study are listed in Table 1. The culture media were Tryptic Soy Broth (TSB), Tryptic Soy Agar (TSA), SOC (2% tryptone, 0.5% yeast extract, 10 mM NaCl, 2.5 mM KCl, 10 mM MgCl_2_, 10 mM MgSO_4_, and 20 mM glucose), Columbia Agar supplemented with 5% (*v*/*v*) of sheep blood (CBA), and Luria–Bertani (LB) medium. When required, the following supplements were added to the different culture media: 10% (*w*/*v*) sucrose (Suc), 30 µg/mL kanamycin (Km), 100 µg/mL spectinomycin (Spc), and 30 µg/mL chloramphenicol (Cm). All bacterial cultures were incubated at 37 °C, under constant shaking at 200 rpm when necessary. The cultures were inoculated with 7 × 10^9^ CFU in a final volume of 20 mL TSB to obtain growth curves. The optical densities were measured at 420 nm every 2 to 4 h.

The construction of *B. abortus ΔbvrRS*. The genes *bvrR* (BAW_12006) and *bvrS* (BAW_12007) of the parental strain *B. abortus* 2308W were mutated using a previously described [24,25], non-polar, unmarked gene excision strategy with modifications (Appendix A). A *bvrR* upstream (Up) fragment from approximately 1 kb to the second codon of the *bvrR* coding sequence (coordinates 2010312 to 2011285) was amplified with the primers bvrRS-Up-For and bvrRS-Up-Rev (Appendix A). A *bvrS* downstream (Dn) fragment containing the last two codons of the *bvrS* coding region to approximately 1 kb downstream (coordinates 2013910 to 2014911) was amplified with the primers bvrRS-Dn-For and bvrRS-Dn-Rev (Appendix A). The Up fragment was cut with *BamHI* (Thermo Fisher Scientific, Vilnius, Lithuania), and the Dn fragment was cut with *PstI* (Thermo Fisher Scientific, Vilnius, Lithuania). Both fragments were treated with polynucleotide kinase in a ligase buffer and included in a single ligation mix with *BamHI*/*PstI*-digested pNPTS138 (Table 1), a suicide vector expressing kanamycin (Km) resistance, and the gene *sacB* for sucrose (Suc) counterselection. The resulting plasmid was called p*ΔbvrRS* (Table 1) and lacked a 2623 bp region located between the Up and Dn fragments in the genome of the parental strain and corresponding to most of the coding sequences of *bvrR* and *bvrS* (coordinates 2011286 to 2013909). Subsequently, 1–3 μL of p*ΔbvrRS* in distilled water was electroporated into 40 μL of *B. abortus* 2308W competent cells prepared as described in [24,25]. Electroporation was performed in a BTX^TM^ cell electroporator using the following parameters: 2.5 kV at 400 ohms and 50 mF. After electroporation, 1 mL of SOC medium was added, and the cells were incubated overnight. Then, 100–200 μL volumes were plated on CBA + Km to select the clones harboring p*ΔbvrRS*. The plates were incubated for 6–10 days. Single colonies were selected, transferred into TSB, and incubated overnight. Volumes of 50, 100, and 200 μL were plated on TSA + 10% Suc to counterselect clones integrating the Up–Dn construct into the chromosome of the parental strain by allelic exchange. The plates were incubated for 3–6 days. Single colonies were picked, and exact replicas were cultured on TSA + 10% Suc and TSA+ Km, to select Suc^R^ and Km^S^ clones that were expected to have lost both *bvrRS* genes and the suicidal plasmid. The plates were incubated for 2–3 days. The Suc^R^ and Km^S^ clones were screened by colony PCR with the primers bvrRS-Con-For and bvrRS-Con-Rev (Appendix A) in a reaction with DreamTaq PCR Master Mix (2X) (Thermo Fisher Scientific, Vilnius, Lithuania). The “Con” primers amplify a 3332 bp fragment in the parental strain (coordinates 2011017 to 2014349) and a 709 bp fragment in the *B. abortus ΔbvrRS* mutant strain. Clones giving the expected result for the mutant strain were also subjected to PCR with the bvrR_bv1 pair of primers (Appendix A). The “bv1” primers amplify a 63 bp fragment of the *bvrR* coding sequence in the parental strain (coordinates 2011468 to 2011531) and are not expected to amplify any PCR product in the mutant strain. Mutant clones selected based on the presence of the 709 bp Con-amplicon and the absence of the 63 bp bv1-amplicon, were subjected to biochemical identification tests and further confirmation by DNA Sanger sequencing, Southern blot, Western blot, and whole-genome sequencing, as described below.

Biochemical identification. Basic biochemical tests were performed as described in [26] using the potential mutant clones and the strain *B. abortus* 2308W. The performed tests included urease (1 h and 24 h), oxidase, H_2_S production, nitrate reduction, and sensitivity to thionine (20 µg/mL) (24 h and 72 h) and basic fuchsin (20 µg/mL) (24 h and 72 h). The presence of a smooth LPS was verified through the acriflavine agglutination test as described in [26].

Southern Blot. The mutation was confirmed by a Southern blot as described in [27], with a few modifications. Genomic DNA from the parental and the mutant strains was extracted with the Wizard^®^ Genomic DNA Purification Kit (Promega, Madison, WI, USA). Purified DNA was double digested with 25 U of *BshTI* (*AgeI*) and 12.5 U of *MlsI* (*MscI*) restriction enzymes (Thermo Scientific, USA), respectively, cutting the *B. abortus* 2308W genomic DNA at positions 2011220 (−60 from *bvrR* start codon) and 2014898 (+983 from *bvrS* stop codon) and, respectively, generating 3679 bp and 1055 bp fragments in the parental strain and in the *B. abortus ∆bvrRS* mutant strains. The digested DNA of each strain was separated through electrophoresis in a 0.6% agarose gel for 2 h at 100 V, and the gel was blotted onto a nylon membrane (Roche, Mannheim, Germany). The probe was generated using the primers bvrRS-South (Appendix A), amplifying a fragment of 299 bp of the coding sequence of the *bvrS* downstream gene (BAW_12008, coordinates 2013977 to 2014276). Probe labeling, hybridization, and detection were performed using the DIG DNA Labeling and Detection kit (Roche, Mannheim, Germany).

Western Blot. The BvrS, BvrR, and Omp25 expressions were assessed as described in [27], with modifications. The bacterial cultures were grown for up to 32 h to test protein expression at different growth phases. The loading control was Omp19.

Sanger sequencing. The Con-amplicon of the mutant strain was Sanger sequenced to confirm the absence of *bvrR* and *bvrS*. The regions recombined during the deletion of the *bvrR* and *bvrS* genes were also sequenced. The upstream recombination region was amplified with *pckA* primers (Appendix A), obtaining an amplicon of 677 bp (coordinates 2009968 to 2010645). The downstream recombination region was amplified with the revtrans20682069-2.3 and revtrans20692070-2.5 primers (Appendix A), obtaining an amplicon of 628 bp (coordinates 2014621 to 2015249). All amplicons were cycle sequenced using the BigDye™ Terminator v3.1 Cycle Sequencing Kit (Thermo Fisher Scientific, Vilnius, Lithuania) and sent to “Centro de Investigación en Biología Celular y Molecular”, at the University of Costa Rica, for capillary electrophoresis. The sequencing results were analyzed using BLAST (Basic Local Alignment Search Tool) [28], with blastn and default parameters to look for their correspondence in the genomic sequence of *B. abortus* 2308, chromosome I (NCBI Reference Sequence: NC_007618.1).

Whole-genome sequencing. For whole-genome sequencing, genomic DNA from *B. abortus* 2308W and *B. abortus ∆bvrRS* was obtained using the Wizard^®^ Genomic DNA Purification Kit (Promega, Madison, WI, USA) from liquid cultures grown to the stationary phase. Genomic libraries were prepared with the rapid barcoding sequencing SQK-RBK004 kit (Oxford Nanopore Technologies, Oxford, UK) for sequencing on a minION platform using the MinKNOW software version 22.08.4, according to the manufacturers’ recommendations. Base calling and the conversion of the raw data to the FASTQ format were performed with Guppy v.3.6.0., while the reads’ quality was verified with nanoplot 1.40.2 [29]. Low-quality reads (Q < 10 and minimum length of 1000) were filtered using nanofilt 2.8.0 [29]. Adapters and barcodes were trimmed with Porechop 0.2.4. Reads that passed the quality check were used for whole-genome assembly with Unicycler v0.4.8 [30]. The assembled genomes were verified with QUAST 5.0.2 (Quality Assessment Tool for Genome Assemblies Version) [31] and CheckM v1.1.3 for completeness and contamination [32]. The project was deposited at DDBJ/ENA/GenBank under the following accession numbers: Bioproject PRJNA891361, assemblies CP109916-CP109917 and CP109914-CP109915, and raw sequencing data SRR21939256 and SRR21939255 for *B. abortus* 2308W and *B. abortus ∆bvrRS*, respectively. The resulting complete-genome assemblies at the chromosome level were compared using BLAST [33] with blastn type and default parameters to verify the genomic differences between *B. abortus ∆bvrRS* and *B. abortus* 2308W. Genome content coverage was also verified by read mapping against the assembled genome using the long-read mapping pipeline Vulcan 1.0.3 [34]. Genomic visualizations were conducted with Artemis Comparison Tool 18.2.0 [35].

Gentamicin-protection assay and intracellular replication quantification. Murine RAW 264.7 macrophages (ATCC TIB-71) or HeLa epithelial cells (ATCC clone CCl-2) were cultivated and infected with *B*. *abortus* strains in the exponential growth phase as described in [5,27]. The number of intracellular, viable *B. abortus* CFUs was determined at 0, 24, and 48 h after infection.

Membrane integrity tests. The sensitivity to the bactericidal action of non-immune serum was tested as described in [27], with an incubation time of 45 min with the non-immune serum instead of 90 min. The minimal inhibitory concentration of polymyxin B in TSB at pH 7.0 and 6.0 was determined by the microdilution method as described in [27].

Statistical analyses. The Kruskal–Wallis test for multiple comparisons was used as described in [27].

## 3. Results

### 3.1. The ∆bvrRS Mutant Strain Is a Null Mutant

In this study, we constructed a null mutant with a deletion in *bvrR* and *bvrS* using an unmarked gene excision approach. The colony PCR screening of the Suc^R^ and Km^S^ clones revealed four potential mutant clones lacking *bvrR* and *bvrS*, numbered 3, 4, 41, and 42 (Appendix A). Clone 4 was selected for further analysis and renamed *B. abortus ∆bvrRS*. This strain tested positive in the following biochemical tests used for the identification of the *Brucella* genus: urease at 1 h and 4 h; oxidase, H_2_S production, and resistance to fuchsin (20 µg/mL) and thionine (20 µg/mL) at 72 h; and negative reactions to nitrate reduction and resistance to fuchsin (20 µg/mL) and thionine (20 µg/mL) at 24 h.

Like the parental strain, *B. abortus ∆bvrRS* excluded crystal violet, did not agglutinate with acriflavine, and agglutinated with antibodies against smooth LPS, indicating the conservation of the S-LPS phenotype as the parental strain. In TSB, *B. abortus ∆bvrRS* consistently exhibited a prolonged lag phase in all independent replicas, retarding entry into the log phase and catching up with *B. abortus* 2308 W in the stationary phase (Figure 1a). The Southern blot results demonstrated a deletion of approximately 3000 bp (Figure 1b); the Western blot results confirmed a lack of expression of BvrR and BvrS (Figure 1c), and the Sanger sequencing results showed an in-frame deletion of the *bvrRS* genes and the correct recombination of the Up and the Dn fragments.

### 3.2. Whole-Genome Sequencing

The obtained coverage was over 68×; the GC content was 57.22% GC, and over 97.99% predicted completeness was achieved with contamination below 0.71% (complete sequencing and assembly results in Appendix A). The genome assembly resulted in two contigs for both strains, representing the complete chromosomes of *B. abortus* composed of 2.1 and 1.2 Mb, respectively.

BLAST comparison between the strain 2308W and the mutant strain confirmed the deletion of the *bvrRS* genes (coordinates 2011286 to 2013909) without further alterations in the genome (Figure 2).

### 3.3. The ∆bvrRS Mutant Strain Displays an Attenuated Phenotype

To characterize the virulence phenotype of *B. abortus ∆bvrRS*, we assessed its ability to replicate intracellularly in HeLa cells and RAW macrophages. As shown in Figure 3, *B. abortus* 2308W was able to replicate intracellularly in both types of cells, but *B. abortus ∆bvrRS* and the transposition mutants, *B. abortus bvrR*::Tn5 and *B. abortus bvrS*::Tn5, did not, confirming attenuation in the cell model.

### 3.4. The ∆bvrRS Mutant Strain Exhibits Altered Membrane Integrity

To assess the membrane integrity of *B. abortus ∆bvrRS*, we first performed a susceptibility to polymyxin B test. As shown in Table 2, at pH 7.0 and pH 6.0, the MIC values of *B. abortus ∆bvrRS* and both transposition mutants, *B. abortus bvrR*::Tn5 and *B. abortus bvrS*::Tn5, were lower by at least two dilutions of difference than those of the parental strain, suggesting altered membrane integrity. All the MICs increased by at least two dilutions of difference compared to that at pH 7.0, indicating a higher resistance under acidic conditions.

Then, we confirmed that *B. abortus ∆bvrRS* was susceptible to the action of a non-immune serum, unlike the parental strain (Figure 4a), and we evaluated the expression of Omp25, known to be directly regulated by the TCS BvrR/BvrS, confirming that *B. abortus ∆bvrRS* displayed a lack of expression of this outer membrane protein, unlike *B. abortus* 2308W (Figure 4b).

## 4. Discussion

The role of the TCS BvrR/BvrS in the virulence of brucellae has been studied in transposition mutants with a kanamycin-resistance marker in the inserted transposon [4]. Construction and characterization of a null mutant was necessary to compare and corroborate phenotypic traits and to obtain an antibiotic-marker-free mutant strain. However, previous attempts to construct clean mutants were unsuccessful, prompting the designation of the *bvrRS* genes as essential [12]. Later it was shown that they did not seem to be essential for extracellular growth [36], which implied that they could be mutated. Since the construction of single mutants in several orthologs of BvrR/BvrS in other cell-associated *Rhizobiales* was consistently reported as difficult [13,14,15,16,17,18,19,20,21], we employed a non-polar, unmarked gene excision strategy [24,25] to construct a *B. abortus* mutant strain with a deletion in both *bvrRS* genes. Different experimental approaches confirmed the in-frame double deletion, and the whole-genome sequencing results demonstrated a clean null mutation without further alterations in the genome.

The phenotypic characterization of *B. abortus ∆bvrRS* revealed a smooth LPS, unaltered biochemical characteristics, retarded lag growth phase defects, inability to replicate intracellularly in cell models, susceptibility to polymyxin B and to human non-immune serum, and lack of expression of Omp25. These results are consistent with previous studies about the role of the TCS BvrR/BvrS that were performed using the transposition mutants *B. abortus bvrR*::Tn5 and *B. abortus bvrS*::Tn5 [4,7,37].

In conclusion, a *B. abortus* null mutant in the *bvrR/bvrS* genes was successfully obtained through allelic replacement. This novel mutant does not harbor any resistant marker gene and will be useful for further studies on the TCS BvrR/BvrS.

## Figures and Tables

**Figure 1 microorganisms-11-02014-f001:**
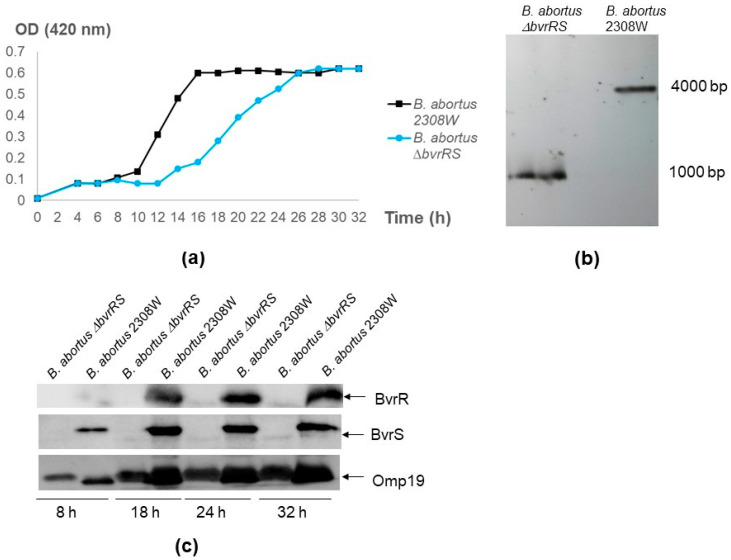
Confirmation of the *B. abortus ∆bvrRS* double-mutant strain. (**a**) Growth curve of *B. abortus ∆bvrRS* compared to *B. abortus* 2308W. An inoculum of 7 × 10^9^ CFU/mL was incubated in TSB at 37 °C and 200 rpm. Optical density (OD) at 420 nm was measured every 2–3 h, up to 32 h. (**b**) Southern blot assay. Genomic DNA of *B. abortus ∆bvrRS* and *B. abortus* 2308W was extracted and digested with *BshTI* (*AgeI*) and *MlsI* (*MscI*). A probe targeting the coding sequence of BAW_12008 downstream of *bvrS*, hybridized with a ~1000 bp band in *B. abortus ∆bvrRS* and a ~4000 bp band in *B. abortus* 2308W. The difference in molecular weight between these two bands matched the size of the excised genes (2635 bp), confirming the deletion of the *bvrRS* genes. (**c**) Western blot analysis of BvrR and BvrS expression according to growth phase in *B. abortus ∆bvrRS* and *B. abortus* 2308W. Both strains were grown in TSB for 32 h, and representative time points of the different growth phases were analyzed according to the growth curves shown in the first panel. Equal amounts (20 μg) of whole-bacterium lysates were separated through 12.5% SDS-PAGE, blotted, and analyzed with anti-BvrR, anti-BvrS, and anti-Omp19 as the loading control. The results are representative of at least three independent experiments.

**Figure 2 microorganisms-11-02014-f002:**
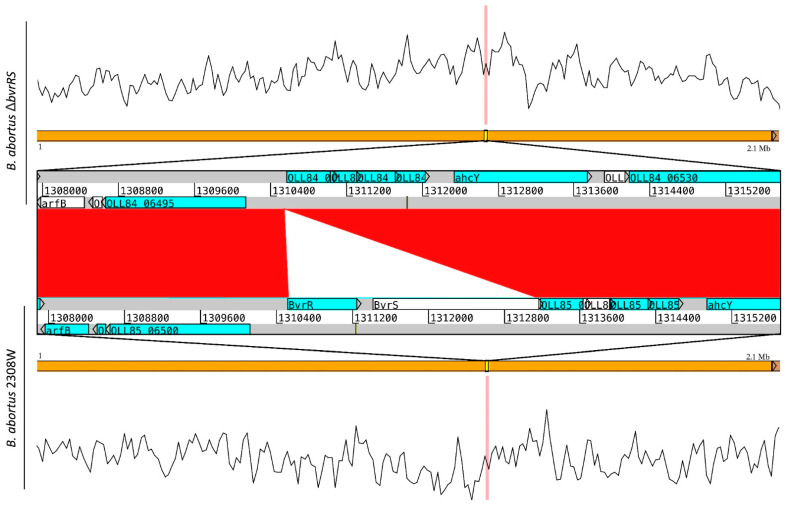
BLAST comparison of the genomic sequences obtained from the strains *B. abortus ∆bvrRS* and *B. abortus* 2308W, confirming the deletion of the *bvrRS* genes in *B. abortus ∆bvrRS*. The orange lines show the 2.1 Mb chromosome of *B. abortus,* with the reads’ coverage across the genome shown by a mapping visualization in the upper and lower graphs. The section of the chromosome with alignment gaps is highlighted and amplified, showing gray lines as the genome sequences and blue boxes as coding sequences (CDSs), and the red in the middle represents 99% identity among regions compared. Deletion of the genes *bvrR* and *bvrS* is shown by the absence of similarity among both regions (white triangle). Up- and downstream CDSs show no additional alterations in *B. abortus ∆bvrRS*.

**Figure 3 microorganisms-11-02014-f003:**
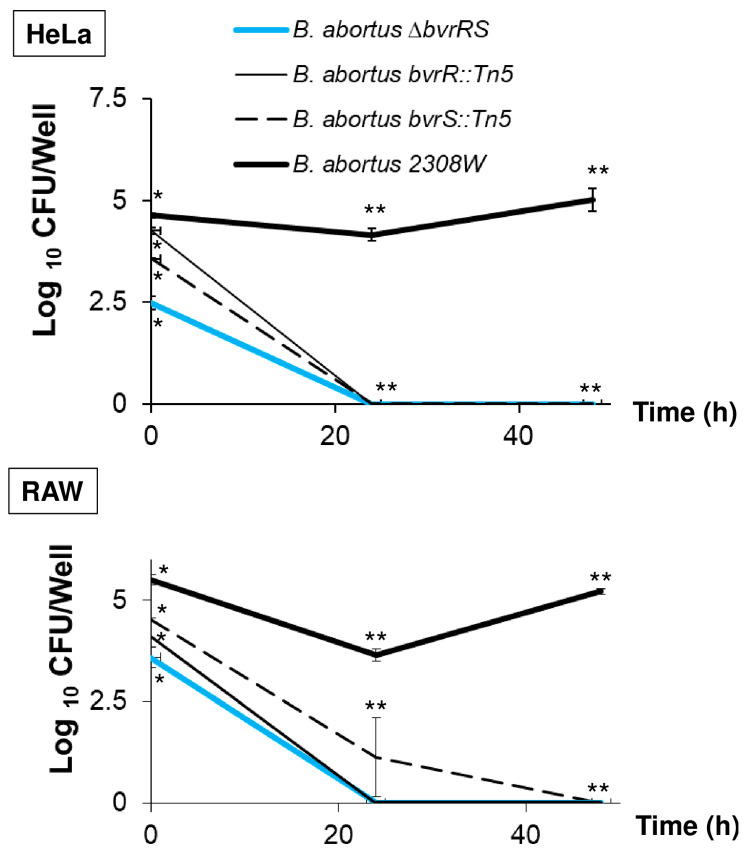
*B. abortus ∆bvrRS* displays lack of replication in HeLa cells and RAW macrophages. Cell cultures were grown in DMEM until approximately 80% of confluence and then inoculated in triplicate with an MOI of 100 for HeLa and 500 for the RAW macrophages. The extracellular bacteria were killed with gentamicin, and the infected cells were incubated for 0, 24, and 48 h, and bacterial counts were performed in triplicate at the three post-infection times. The results are representative of at least three independent experiments. * Statistically significant differences (*p* < 0.05) between the four strains (Kruskal–Wallis test for multiple comparisons). ** Statistically significant differences (*p* < 0.05) between *B. abortus* 2308W and the other three strains (Kruskal–Wallis test for multiple comparisons).

**Figure 4 microorganisms-11-02014-f004:**
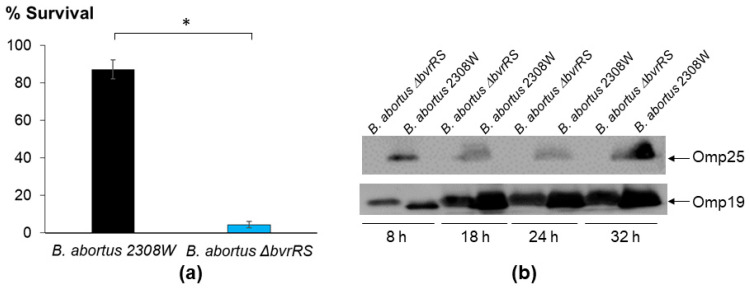
Characterization of the membrane integrity of *B. abortus ∆bvrRS*. (**a**) Loss of resistance to non-immune serum in *B. abortus ∆bvrRS* compared to *B. abortus* 2308W. Both strains were exposed to non-immune human serum at 37 °C for 45 min, and the survival percentage was calculated. (**b**) Western blot analysis of Omp25 expression, according to growth phase in *B. abortus ∆bvrRS* and *B. abortus* 2308W. Both strains were grown in TSB for 32 h, and representative time points of the different growth phases were analyzed according to the growth curves shown in Figure 1a. Equal amounts (20 μg) of whole-bacterium lysates were separated through 12.5% SDS-PAGE, blotted, and analyzed with anti-Omp25 and Omp19 as the loading control. These results are representative of at least three independent experiments. * *p* < 0.05 (Kruskal–Wallis test for multiple comparisons).

**Table 1 microorganisms-11-02014-t001:** Bacterial strains and plasmids used in this study.

Bacterial Strains/Plasmids	Phenotype/Characteristics	Source/Reference
*B. abortus* 2308W	Parental strain, smooth LPS, Nal^R^, virulent	[23]
*B. abortus bvrS*::Tn5	2308-derivative, *bvrS*::Tn5, smooth LPS, attenuated	[4]
*B. abortus bvrR*::Tn5	2308-derivative, *bvrR*::Tn5, smooth LPS, attenuated	[4]
*B. abortus ΔbvrRS*	2308W-derivative, *ΔbvrRS*, smooth LPS, attenuated	This study
pNTPS138	Suicide vector, *oriT*, *sacB* Km^R^	M.R.K. Alley, unpublished
p*ΔbvrRS*	pNTPS138 derivative, *ΔbvrRS*, Km^R^	Courtesy of Clayton C. Caswell

**Table 2 microorganisms-11-02014-t002:** Results of the susceptibility test for polymyxin B.

Strain	MIC (µg/mL)	
	pH 7.0	pH 6.0
*B. abortus* 2308W	4–8	64
*B. abortus ΔbvrRS*	1–4	4–8
*B. abortus bvrR*::Tn5	2	8–16
*B. abortus bvrS*::Tn5	2	16

## Data Availability

The data presented in this study are openly available in GenBank, accession numbers [CP109914, CP109915, CP109916, CP109917], and the SRA accession numbers for the raw reads are SRR21939256 and SRR21939255.

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
