# Peer review of "A bvrR/bvrS Non-Polar Brucella abortus Mutant Confirms the Role of the Two-Component System BvrR/BvrS in Virulence and Membrane Integrity"

_microorganisms, 2023, doi:10.3390/microorganisms11082014_

Round 1

Reviewer 1 Report

The manuscript by Olga Rivas-Solano describes the construction of a B. abortus mutant strain lacking both genes of the TCS BvrR/BvrS using an unmarked gene excision strategy, to confirm the role of BvrR/BvrS in pathogenesis and membrane integrity.

Overall, this manuscript is written exceptionally and very well structured. It shows a deep understanding of the literature and the gaps in the field. All the experiments are very well described, and the discussion ties the whole thing together.

I really enjoyed reading this manuscript, and I have no further comments to add.

Author Response

Dear Reviewer,

We would like to thank you for the overall positive comments on the manuscript by Rivas-Solano et al. entitled: “A bvrR/bvrS non-polar Brucella abortus mutant confirms the role of the two-component system BvrR/BvrS in virulence and membrane integrity”.

We also would like to inform you that the Editor requested minor changes to increase the word count from 3935 to a minimum of 4000 words. Therefore, we extended the introduction section to better describe the symptoms of brucellosis in animals and humans. In a separated paragraph, we related the brucellae intracellular trafficking to a highly coordinated gene expression (mainly through two-component systems). We did not need to add any additional references for those changes. The lines 35 to 41 of the original manuscript, stating:

Brucella organisms are Gram-negative facultative extracellular-intracellular zoonotic pathogens. Brucella abortus causes abortion and infertility in cattle and a chronic debilitating illness in humans [1]. The virulence of brucellae depends on their ability to invade, survive, and replicate inside their host cells [2,3], which requires coordinated gene expression. In brucellae, the two-component signal transduction system (TCS) BvrR/BvrS comprises the sensor membrane protein BvrS and its cognate cytoplasmic response regulator BvrR.”

Were modified as follows in the revised version (lines 35 to 52):

Brucella organisms are Gram-negative facultative extracellular-intracellular pathogens, causing brucellosis, a neglected zoonotic disease distributed worldwide. Brucella abortus infects cattle and exhibits a strong tropism for the reproductive system, causing abortion, infertility, decreased milk production, reproductive failure, epididymitis, and hence, economic losses. Humans are accidental hosts initially developing an acute febrile disease that could become a chronic infection with osteoarticular, gastrointestinal, hepatobiliary, pulmonary, genitourinary, cardiovascular and neurological complications[1].

Brucellae lack classical virulence factors and their pathogenicity depends on their ability to invade, survive, and replicate inside host professional and non-professional phagocytes. During their intracellular lifecycle, brucellae avoid the endocytic pathway and redirect their trafficking to a replicative compartment derived from the endoplasmic reticulum [2,3]. Therefore, the adaptation to this intracellular trafficking requires an extremely well-coordinated gene expression, mainly through two-component signal transduction systems (TCS).

In brucellae, the TCS BvrR/BvrS comprises the sensor membrane protein BvrS and its cognate cytoplasmic response regulator BvrR.”

Best regards,

Prof. Olga Rivas-Solano.

Centro de Investigación en Biotecnología

Instituto Tecnológico de Costa Rica

Reviewer 2 Report

Well written study with clear methods and data presentation. Biologically this study does not add much to the literature, but it does establish a road map for knocking out genes in Brucella using a targeted, non-antibiotic dependent approach. 

The question addressed by this report is whether a targeted genetic deletion in the 2-component BvrR/BvrS system presents the same phenotype as the previously generated transposon mutants. Genetic deletion mutants have also been difficult to generate in Brucella, so this work establishes a roadmap for other researchers in the field. Whereas the technical approach could be helpful, the information gained is extremely incremental and adds little to our understanding of Brucella. As stated, the researcher’s goals have been met. The methods are sound and presentation clear. Conclusions are consistent with the data. References are appropriate. 

Author Response

(The authors gave the same response as above.)

Reviewer 3 Report

In the article, Olga Rivas-Solano and her collaborators describe the generation of a mutant strain of B. abortus with deletions in both bvrR and bvrS genes. Through various assays using the mutant strain, they were able to confirm the significance of the two-component regulatory system BvrR/BvrS and validated the previous characterizations using attenuated transposition mutants bvrR::Tn5 and bvrS::Tn5.

The article is well-written, the methodology is appropriate, and the categorization of the mutated strain is comprehensive. The B. abortus ∆bvrRS strain, lacking an antimicrobial marker, will enable trials aimed at determining the importance of BvrR/BvrS in the brucella life cycle.                                        

Author Response

(The authors gave the same response as above.)
